# GLAMbox: A Python toolbox for investigating the association between gaze allocation and decision behaviour

Felix Molter[1,2,3,4☯]*, Armin W. Thomas[2,3,5,6☯]*, Hauke R. Heekeren[2,3], Peter N. C. Mohr[1,2,4]

**1** WZB Berlin Social Science Center, Berlin, Germany, **2** Center for Cognitive Neuroscience Berlin, Freie Universität Berlin, Berlin, Germany, **3** Department of Education and Psychology, Freie Universität Berlin, Germany, **4** School of Business and Economics, Freie Universität Berlin, Germany, **5** Department of Electrical Engineering and Computer Science, Technische Universität Berlin, Berlin, Germany, **6** Max Planck School of Cognition, Leipzig, Germany

☯ These authors contributed equally to this work.
* felixmolter@gmail.com (FM); athms.research@gmail.com (AT)

**Data Availability Statement:** All files are available at Github under https://github.com/glamlab/glambox.

## Abstract

Recent empirical findings have indicated that gaze allocation plays a crucial role in simple decision behaviour. Many of these findings point towards an influence of gaze allocation onto the speed of evidence accumulation in an accumulation-to-bound decision process (resulting in generally higher choice probabilities for items that have been looked at longer). Further, researchers have shown that the strength of the association between gaze and choice behaviour is highly variable between individuals, encouraging future work to study this association on the individual level. However, few decision models exist that enable a straightforward characterization of the gaze-choice association at the individual level, due to the high cost of developing and implementing them. The model space is particularly scarce for choice sets with more than two choice alternatives. Here, we present GLAMbox, a Python-based toolbox that is built upon PyMC3 and allows the easy application of the gaze-weighted linear accumulator model (GLAM) to experimental choice data. The GLAM assumes gaze-dependent evidence accumulation in a linear stochastic race that extends to decision scenarios with many choice alternatives. GLAMbox enables Bayesian parameter estimation of the GLAM for individual, pooled or hierarchical models, provides an easy-to-use interface to predict choice behaviour and visualize choice data, and benefits from all of PyMC3's Bayesian statistical modeling functionality. Further documentation, resources and the toolbox itself are available at https://glambox.readthedocs.io.

## Introduction

A plethora of empirical findings has established an association between gaze allocation and decision behaviour on the group-level. For example, in value-based decision making, it has been repeatedly shown that longer gaze towards one option is associated with a higher choice

**Funding:** The Junior Professorship of P.N.C.M. as well as the associated Dahlem International Network Junior Research Group Neuroeconomics is supported by Freie Universität Berlin within the Excellence Initiative of the German Research Foundation (DFG). Further support for P.N.C.M. is provided by the WZB Berlin Social Science Center. F.M. is supported by the International Max Planck Research School on the Life Course (LIFE). A.T. is supported by the BMBF and Max Planck Society. The funders had no role in study design, data collection and analysis, decision to publish, or preparation of the manuscript.

**Competing interests:** The authors have declared that no competing interests exist.

probability for that option [1–13] and that external manipulation of gaze allocation changes choice probabilities accordingly [1, 9, 10, 14]. Such gaze bias effects are not limited to value-based decisions, but have recently also been observed in perceptual choices, where participants judge the perceptual attributes of stimuli based on available sensory information [14].

These findings have led to the development of a set of computational models, aimed at capturing the empirically observed association between gaze allocation and choice behaviour by utilizing gaze data to inform the momentary accumulation rates of diffusion decision processes [2, 7, 8, 14–17]. Specifically, these models assume that evidence accumulation in favour of an item continues while it is not looked at, but at a discounted rate. The application of these models is limited so far, as fitting them to empirical data depends on computationally expensive simulations, involving the simulation of fixation trajectories. These simulations, as well as the creation of models of the underlying fixation process, become increasingly difficult with increasing complexity of the decision setting (e.g., growing choice set sizes or number of option attributes, etc). Existing approaches that circumvent the need for simulations, model the evidence accumulation process as a single diffusion process between two decision bounds and are therefore limited to binary decisions [2, 18].

However, researchers are increasingly interested in choice settings involving more than two alternatives. Choices outside the laboratory usually involve larger choice sets or describe items on multiple attributes. Besides, many established behavioural effects only occur in multi-alternative and multi-attribute choice situations [19].

Furthermore, recent findings indicate strong individual differences in the association between gaze allocation and choice behaviour [20, 21] as well as individual differences in the decision mechanisms used [15]. While the nature of individual differences in gaze biases is still not fully understood, different mechanisms have been suggested: Smith and Krajbich [20] showed that gaze bias differences can be related to individual differences in attentional scope ("tunnel vision"). Vaidya and Fellows [13] found stronger gaze biases in patients with damage in dorsomedial prefrontal cortex (PFC). Further, recent empirical work has investigated the roles of learning and attitude accessibility in gaze dependent decision making [22, 23]. However, more systematic investigations of these differences are needed, as the majority of model-based investigations of the relationship between gaze allocation and choice behaviour were focused on the group level, disregarding differences between individuals.

With the Gaze-weighted linear accumulator model (GLAM; [21]), we have proposed an analytical tool that allows the model-based investigation of the relationship between gaze allocation and choice behaviour at the level of the individual, in choice situations involving more than two alternatives, solely requiring participants' choice, response time (RT) and gaze data, in addition to estimates of the items' values.

Like the attentional Drift Diffusion Model (aDDM) [7, 8, 17], the GLAM assumes that the decision process is biased by momentary gaze behaviour: While an item is not fixated, its value representation is discounted. The GLAM, however, differs from the aDDM in other important aspects: In contrast to the aDDM, the fixation-dependent value signals are averaged across the trial, using the relative amount of time individuals spend fixating the items. This step abstracts away the specific sequence of fixations in a trial, that can be investigated with the aDDM. On the other hand, this simplification allows for the construction of trial-wise constant drift rates that can enter a basic stochastic race framework. While race models like the GLAM are not statistically optimal [24] the GLAM has been shown to provide a good fit to empirical data [21]. In general, race models have at least two practical advantages: First, they often have analytical solutions to their first-passage density distributions, and secondly, they naturally generalize to choice scenarios involving more than two alternatives. The analytical tractability of the race

framework further allows for efficient parameter estimation in a hierarchical Bayesian manner. The GLAM thereby integrates gaze-dependent accumulation into a practical race model shell.

To make GLAM more accessible, we now introduce GLAMbox, a Python-based toolbox for the application of the GLAM to empirical choice, RT and gaze data. GLAMbox allows for individual and hierarchical estimation of the GLAM parameters, simulation of response data and model-based comparisons between experimental conditions and groups. It further contains a set of visualization functions to inspect choice and gaze data and evaluate model fit. We illustrate three application examples of the toolbox: In Example 1, we illustrate how GLAMbox can be used to analyze individual participant data with the GLAM. In particular, we perform an exemplary model comparison between multiple model variants on the individual level, as well as an out-of-sample prediction of participants' choice and RT data. In Example 2, we demonstrate the application of the GLAM to perform a comparison of group-level parameters in a setting with limited amounts of data, using hierarchical parameter estimation. Lastly, in Example 3, we walk the reader through a step-by-step parameter recovery study with the GLAM, which is encouraged to increase confidence in the estimated parameter values.

## Materials and methods

### Gaze-weighted linear accumulator model details

Like the aDDM, the GLAM assumes that preference formation, during a simple choice process, is guided by the allocation of visual gaze (for an overview, see Fig 1). Particularly, the decision process is guided by a set of decision signals: An absolute and relative decision signal. Throughout the trial, the absolute signal of an item $i$ can be in two states: An unbiased state,

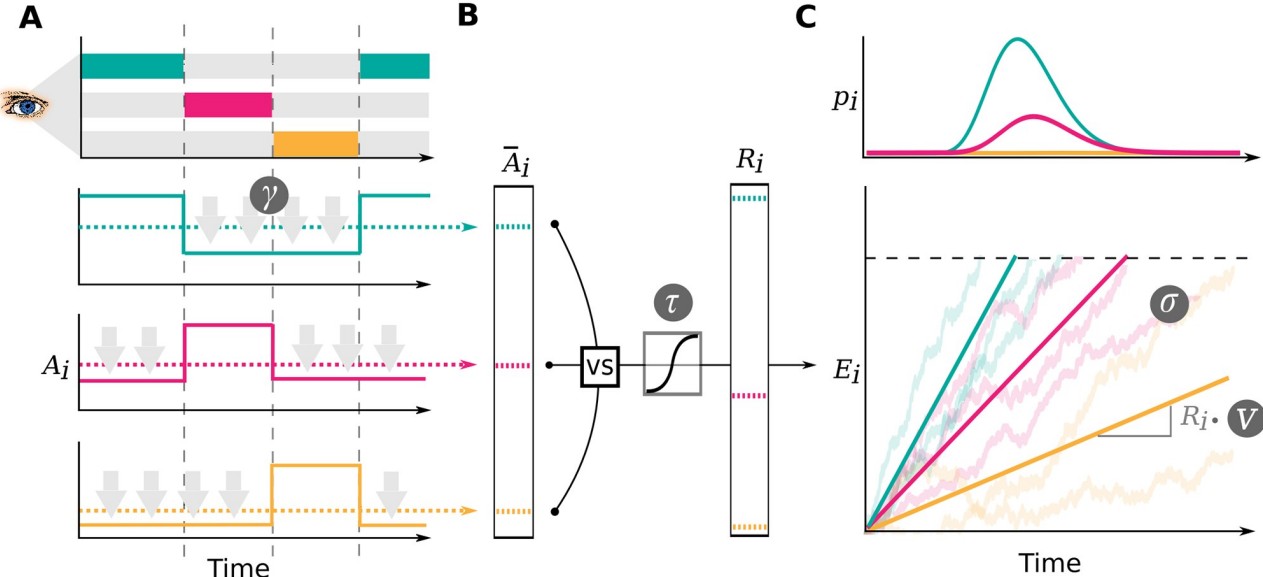

**Fig 1. Gaze-weighted linear accumulator model.** In the GLAM, preference formation during the decision process is dependent on the allocation of visual gaze (A). For each item in the choice set, an average absolute decision signal $\bar{A}_i$ is computed (dashed lines in A). The magnitude of this signal is determined by the momentary allocation of visual gaze: While an item is currently not looked at, its signal is discounted by parameter $\gamma$ ($\gamma \leq 1$; discounting is illustrated by gray arrows) (A). To determine a relative decision signal $R_i$ for each item in the choice set, absolute evidence signals are transformed in two steps (B): First, the difference between each average absolute decision signal $\bar{A}_i$ and the maximum of all others is determined. Second, the resulting differences are scaled through a logistic transform, as the GLAM assumes an adaptive representation of the relative decision signals that is especially sensitive to differences close to 0 (where the absolute signal for an item is very close to the maximum of all others). The resulting relative decision signals $R_i$ can be used to predict choice and RT, by determining the speed of the accumulation process in a linear stochastic race (C). The stochastic race then provides first-passage time distributions $p_i$, describing the likelihood of each item being chosen at each time point.

equal to the item's value $r_i$ while the item is looked at, and a biased state while any other item is looked at, where the item value $r_i$ is discounted by a parameter $\gamma$. The average absolute decision signal $\bar{A}_i$ is given by

$$\bar{A}_i = g_i r_i + (1 - g_i)\gamma r_i, \tag{1}$$

where $g_i$ is defined as the fraction of total trial time that item $i$ was looked at. If $\gamma = 1$, there is no difference between the biased and unbiased state, resulting in no influence of gaze allocation on choice behaviour. For $\gamma$ values less than 1, the absolute decision signal $A_i$ is discounted, resulting in generally higher choice probabilities for items that have been looked at longer. For $\gamma$ values less than 0, the sign of the absolute decision signal $A_i$ changes, when the item is not looked at, leading to an overall even stronger gaze bias, as evidence for these items is actively lost, when they are not looked at. This type of gaze-dependent leakage mechanism is supported by a variety of recent empirical findings [15, 21].

To determine the relative decision signals, the average absolute decision signals $\bar{A}_i$ are transformed in two steps: First, for each item $i$, the relative evidence $R_i^*$ is computed as the difference between the average absolute decision signal of the item $\bar{A}_i$ (Eq 1) and the maximum of all other average absolute decision signals $\bar{A}_{j \neq i}$ (also obtained from Eq 2) is computed:

$$R_i^* = \bar{A}_i - \max_{j \neq i} \bar{A}_j. \tag{2}$$

Second, the resulting difference signals $R_i^*$ are scaled through a logistic transform $s(x)$. The GLAM assumes an adaptive representation of the relative decision signals, which is maximally sensitive to small differences in the absolute decision signals close to 0 (where the difference between the absolute decision signal of an item and the maximum of all others is small):

$$R_i = s(R_i^*) \tag{3}$$

$$s(x) = \frac{1}{1 + \exp(-\tau x)} \tag{4}$$

The sensitivity of this transform is determined by the temperature parameter $\tau$ of the logistic function. Larger values of $\tau$ indicate stronger sensitivity to small differences in the average absolute decision signals $\bar{A}_i$.

Unlike more traditional diffusion models (including the aDDM), the GLAM employs a linear stochastic race to capture response behaviour as well as RTs. The relative signals $R_i$ enter a race process, where one item accumulator $E_i$ is defined for each item in the choice set:

$$E_i(t) = E_i(t-1) + \nu R_i + N(0, \sigma^2), \text{ with } E_i(0) = 0 \tag{5}$$

At each time step $t$, the amount of accumulated evidence is determined by the accumulation rate $\nu R_i$, and zero-centered normally distributed noise with standard deviation $\sigma$. The velocity parameter $\nu$ linearly scales the item drift rates in the race process and thereby affects the response times produced by the model: Lower values of $\nu$ produce longer response times, larger $\nu$s result in shorter response times. A choice for an item is made as soon as one accumulator reaches the decision boundary $b$. To avoid underdetermination of the model, either the velocity parameter $\nu$, the noise parameter $\sigma$ or the boundary has to be fixed. Similar to the aDDM, the GLAM fixes the boundary to a value of 1. The first passage time density $f_i(t)$ of a single linear stochastic accumulator $E_i$, with decision boundary $b$, is given by the inverse

Gaussian distribution:

$$f_i(t) = \left[\frac{\lambda}{2\pi t^3}\right]^{\frac{1}{2}} \exp\left(\frac{-\lambda(t-\mu)^2}{2\mu^2 t}\right) \tag{6}$$

$$\text{with } \mu = \frac{b}{vR_i} \text{ and } \lambda = \frac{b^2}{\sigma^2}$$

However, this density does not take into account that there are multiple accumulators in each trial racing towards the same boundary. For this reason, $f_i(t)$ must be corrected for the probability that any other accumulator crosses the boundary first. The probability that an accumulator crosses the boundary prior to $t$, is given by its cumulative distribution function $F_i(t)$:

$$F_i(t) = \Phi\left(\sqrt{\frac{\lambda}{t}}\left(\frac{t}{\mu} - 1\right)\right) + \exp\left(\frac{2\lambda}{\mu}\right) \cdot \Phi\left(-\sqrt{\frac{\lambda}{t}}\left(\frac{t}{\mu} + 1\right)\right) \tag{7}$$

Here, $\Phi(x)$ defines the standard normal cumulative distribution function. Hence, the joint probability $p_i(t)$ that accumulator $E_i$ crosses $b$ at time $t$, and that no other accumulator $E_{j \neq i}$ has reached $b$ first, is given by:

$$p_i(t) = f_i(t) \prod_{j \neq i} (1 - F_j(t)) \tag{8}$$

**Contaminant response model.** To reduce the influence of erroneous responses (e.g., when the participant presses a button by accident or has a lapse of attention during the task) on parameter estimation, we include a model of contaminant response processes in all estimation procedures: In line with existing drift diffusion modelling toolboxes [25], we assume a fixed 5% rate of erroneous responses $\epsilon$ that is modeled as a participant-specific uniform likelihood distribution $u_s(t)$. This likelihood describes the probability of a random choice for any of the $N$ available choice items at a random time point in the interval of empirically observed RTs [25, 26]:

$$u_s(t) = \frac{1}{N(\max \text{rt}_s - \min \text{rt}_s)} \tag{9}$$

The resulting likelihood for participant $s$ choosing item $i$, accounting for erroneous responses, is then given by:

$$l_i(t) = (1 - \epsilon) \cdot p_i(t) + \epsilon \cdot u_s(t) \tag{10}$$

The rate of error responses $\epsilon$ can be specified by the user to a different value than the default of 5% using the `error_weight` keyword in the `make_model` method (see below).

**Individual parameter estimation details.** The GLAM is implemented in a Bayesian framework using the Python library PyMC3 [27]. The model has four parameters ($v$, $\gamma$, $\sigma$, $\tau$). By default, uninformative, uniform priors between sensible limits (derived from earlier

applications to four different datasets: [21]) are placed on all parameters:

$$
\begin{aligned}
\nu &\sim U(0,4) \\
\gamma &\sim U(-2,1) \\
\sigma &\sim U(0,4) \\
\tau &\sim U(0,10)
\end{aligned}
$$

These limits were derived by extending the range of observed parameter estimates in earlier applications of the GLAM to four different empirical choice datasets. These datasets encompass data of 117 participants in value-based and perceptual choice tasks with up to three choice alternatives (including a wide range of possible response times, gaze bias strengths and choice accuracies; for further details [21]). Parameter estimates for these datasets are illustrated and summarised in S1 Table, S1 Fig and S1 Fig.

The velocity parameter $\nu$ and the noise parameter $\sigma$ must be strictly positive. Smaller $\nu$ produce slower and less accurate responses (for constant $\sigma$), while smaller $\sigma$ produce more accurate and slower responses (for constant $\nu$). The gaze bias parameter $\gamma$ has a natural upper bound at 1 (indicating no gaze bias), while decreasing $\gamma$ values indicate an increasing gaze bias strength. The sensitivity parameter $\tau$ has a natural lower bound at 0 (resulting in no sensitivity to differences in average absolute decision signals $\bar{A}_i$), with larger values indicating increased sensitivity.

**Hierarchical parameter estimation details.** For hierarchical models, individual parameters are assumed to be drawn from Truncated Normal distributions, parameterized by mean and standard deviation, over which weakly informative, Truncated Normal priors are assumed (based on the distribution of group level parameter estimates obtained from four different datasets in [21]; see Fig 2, S1 and S2 Figs and S1 Table):

$$
\begin{aligned}
\nu_{\mu} &\sim N(0.63, 10 \cdot 0.26), \text{ truncated to } [0,2] \\
\nu_{\sigma} &\sim N(0.26, 10 \cdot 0.11), \text{ truncated to } [0,1] \\
\gamma_{\mu} &\sim N(0.12, 10 \cdot 0.11), \text{ truncated to } [-2,1] \\
\gamma_{\sigma} &\sim N(0.35, 10 \cdot 0.1), \text{ truncated to } [0,1] \\
\sigma_{\mu} &\sim N(0.27, 10 \cdot 0.08), \text{ truncated to } [0,1] \\
\sigma_{\sigma} &\sim N(0.05, 10 \cdot 0.01), \text{ truncated to } [0,0.2] \\
\tau_{\mu} &\sim N(1.03, 10 \cdot 0.58), \text{ truncated to } [0,5] \\
\tau_{\sigma} &\sim N(0.62, 10 \cdot 0.26), \text{ truncated to } [0,3]
\end{aligned}
$$

## Basic usage

**Data format, the `GLAM` class.** The core functionality of the GLAMbox is implemented in the `GLAM` model class. To apply the GLAM to data, an instance of the model class needs to be instantiated and supplied with the experimental data, first:

```
import glambox as gb
glam = gb.GLAM(data=data)
```

The data must be a pandas [28] DataFrame with one row per trial, containing the following variable entries:

- `subject`: Subject index (integer, first subject should be 0)

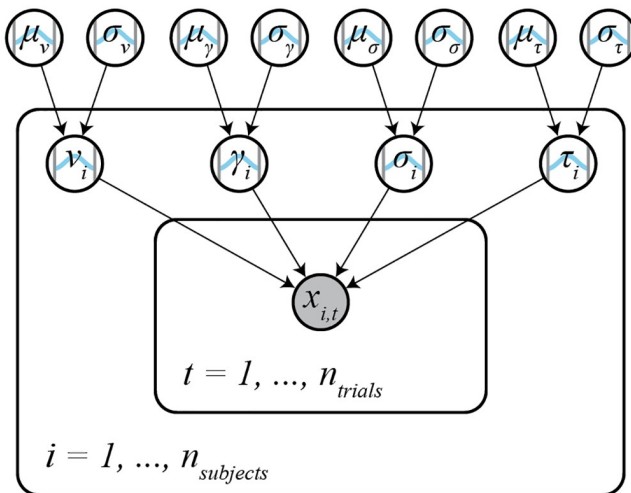

**Fig 2. Hierarchical model structure.** In the hierarchical model, individual subject parameters $\gamma_i$, $v_i$. $\sigma_i$, and $\tau_i$ (subject plate) are drawn from Truncated Normal group level distributions with means $\mu$ and standard deviations $\sigma$ (outside of the subject plate). Weakly informative Truncated Normal priors are placed on the group level parameters. RT and choice data $x_{i,t}$ for each trial $t$ is distributed according to the subject parameters and the GLAM likelihood (Eq (8); inner trial plate).

- `trial`: Trial index (integer, first trial should be 0)

- `choice`: Chosen item in this trial (integer, items should be 0, 1, . . ., $N$)

- `rt`: Response time (float, in seconds)

- for each item $i$ in the choice set:

  - `item_value_i`: The item value (float, we recommend to re-scale all item values to a range between 1 and 10 to allow comparison of parameter estimates between studies)

  - `gaze_i`: The fraction of total time in this trial that the participant spent looking at this item (float, between 0 and 1)

- additional variables coding groups or conditions (string or integer)

For reference, the first two rows of a pandas DataFrame ready to be used with GLAMbox are shown in Table 1.

Next, the respective PyMC3 model, which will later be used to estimate the model's parameters, can be built using the `make_model` method. Here, the researcher specifies the kind of the model: '*individual*' if the parameters should be estimated for each subject individually, '*hierarchical*' for hierarchical parameter estimation, or '*pooled*' to estimate a single parameter set for all subjects. At this stage, the researcher can also specify experimental parameter dependencies: For example, a parameter could be expected to vary between groups or conditions. In line with existing modeling toolboxes (e.g., [25, 29]) dependencies are

**Table 1. The first two rows of a pandas DataFrame ready to be used with GLAM.**

| subject | trial | choice | rt | item_value_0 | item_value_1 | item_value_2 | gaze_0 | gaze_1 | gaze_2 | speed |
|---------|-------|--------|-------|--------------|--------------|--------------|--------|--------|--------|-------|
| 0 | 0 | 0 | 2.056 | 5 | 1 | 3 | 0.16 | 0.62 | 0.22 | 'fast' |
| 0 | 1 | 2 | 3.685 | 3 | 6 | 9 | 0.44 | 0.22 | 0.34 | 'slow' |

defined using the `depends_on` argument. `depends_on` expects a dictionary with parameters as keys and experimental factors as values (e.g., `depends_on=dict(v='speed')` for factor '`speed`' with conditions '`fast`' and '`slow`' in the data). The toolbox internally handles within- and between subject designs and assigns parameters accordingly. If multiple conditions are given for a factor, one parameter will be designated for each condition. Finally, the `make_model` method allows parameters to be fixed to a specific value using the `*_val` arguments (e.g., `gamma_val=1` for a model without gaze bias). If parameters should be fixed for individual subjects, a list of individual values needs to be passed.

```
model.make_model(kind='individual',
                 depends_on=dict(v='speed'),
                 gamma_val=1)
```

**Inference.**   Once the PyMC3 model is built, parameters can be estimated using the `fit` method:

```
model.fit(method='MCMC')
```

The `fit` method defaults to Markov-Chain-Monte-Carlo (MCMC; [30]) sampling, but also allows for Variational Inference (see below).

**Markov-Chain-Monte-Carlo.**   MCMC methods approximate the Bayesian posterior parameter distributions, describing the probability of a parameter taking certain values given the data and prior probabilities, through repeated sampling. GLAMbox can utilize all available MCMC step methods provided by PyMC3. The resulting MCMC traces can be accessed using the `trace` attribute of the model instance (note that a list of traces is stored for models of kind '`individual`'). They should always be checked for convergence, to ascertain that the posterior distribution is approximated well. Both qualitative visual and more quantitative numerical checks of convergence, such as the Gelman-Rubin statistic $\hat{R}$ and the number of effective samples are recommended (for detailed recommendations, see [31, 32]). PyMC3 contains a range of diagnostic tools to perform such checks (such as the `summary` function).

**Variational inference.**   Estimation can also be done using all other estimation procedures provided in the PyMC3 library. This includes variational methods like Automatic Differentiation Variational Inference (ADVI; [33]). To use variational inference, the `method` argument can be set to '`VI`', defaulting to the default variational method in PyMC3. We found variational methods to quickly yield usable, but sometimes inaccurate parameter estimates, and therefore recommend using MCMC for final analyses.

**Accessing parameter estimates.**   After parameter estimation is completed, the resulting estimates can be accessed with the `estimates` attribute of the GLAM model instance. This returns a table with one row for each set of parameter estimates for each individual and condition in the data. For each parameter, a *maximum a posteriori* (MAP) estimate is given, in addition to the 95% Highest-Posterior Density Interval (HPD). If the parameters were estimated hierarchically, the table also contains estimates of the group-level parameters.

**Comparing parameters between groups or conditions.**   Parameter estimates can be compared between different experimental groups or conditions (specified with the `depends_on` keyword when calling `make_model`) using the `compare_parameters` function from the `analysis` module. It takes as input the fitted GLAM instance, a list of parameters ('`v`', '`s`', '`gamma`', '`tau`'), and a list of pairwise comparisons between groups or conditions. The comparison argument expects a list of tuples (e.g., `[('group1', 'group2')`, `('group1', 'group3')]`). For example, given a fitted model instance (here `glam`) a comparison of the $\gamma$ parameter between two groups (`group1` and `group2`) can be computed as:

```
from gb.analysis import compare_parameters
comparison = compare_parameters(model=glam,
                                parameters=['gamma'],
                                comparisons=[('group1', 'group2')])
```

The function then returns a table with one row per specified comparison, and columns containing the mean posterior difference, percentage of the posterior above zero, and corresponding 95% HPD interval. If supplied with a hierarchical model, the function computes differences between group-level parameters. If an individual type model is given, it returns comparison statistics for each individual.

Comparisons can be visualized using the `compare_parameters` function from the `plots` module. It takes the same input as its analogue in the `alysis` module. It plots posterior distributions of parameters and the posterior distributions of any differences specified using the `comparisons` argument. For a usage example and plot see Example 2.

**Comparing model variants.**   Model comparisons between multiple GLAM variants (e.g., full and restricted variants) can be performed using the `compare_models` function, which wraps the function of the same name from the PyMC3 library. The `compare_models` function takes as input a list of fitted model instances that are to be compared. Additional keyword arguments can be given and are passed on to the underlying PyMC3 `compare` function. This allows the user, for example, to specify the information criterion used for the comparison via the `ic` argument (`'WAIC'` or `'LOO'` for Leave-One-Out cross validation). It returns a table containing an estimate of the specified information criterion, standard errors, difference to the best-fitting model, standard error of the difference, and other output variables from PyMC3 for each inputted model (and subject, if individually estimated models were given). We refer the reader to Example 1 for a usage example and exemplary output from the `compare_models` function.

**Predicting choices and response times.**   Choices and RTs can be predicted with the GLAM by the use of the `predict` method:

```
model.predict(n_repeats=50)
```

For each trial of the dataset that is attached to the model instance, this method predicts a choice and RT according to Eq (10), using the previously determined MAP parameter estimates. To obtain a stable estimate of the GLAM's predictions, as well as the noise contained within them, it is recommended to repeat every trial multiple times during the prediction. The number of trial repeats can be specified with the `n_repeats` argument. After the prediction is completed, the predicted data can be accessed with the `prediction` attribute of the model.

## Results

### Example 1: Individual level data & model comparison

Our first example is based on the study by [21]. Here, the authors study the association between gaze allocation and choice behaviour on the level of the individual. In particular, they explore whether (1) gaze biases are present on the individual level and (2) the strength of this association varies between individuals. In this example, we replicate this type of individual model-based analysis, including parameter estimation, comparison between multiple model variants, and out-of-sample prediction of choice and RT data.

**Simulating data.**   First, we simulate a dataset containing 30 subjects, each performing 300 simple value-based choice trials. We assume that in each trial participants are asked to choose the item that they like most out of a set of three presented alternatives (e.g., snack food items;

similar to the task described in [8]). While participants perform the task, their eye movements, choices and RTs are measured. Before completing the choice trials, participants were asked to indicate their liking rating for each of the items used in the choice task on a liking rating scale between 1 and 10 (with 10 indicating strong liking and 1 indicating little liking). The resulting dataset contains a liking value for each item in a trial, the participants' choice and RT, as well as the participant's gaze towards each item in a trial (describing the fraction of trial time that the participant spent looking at each item in the choice set).

To simulate individuals' response behaviour, we utilize the parameter estimates that were obtained by [21] for the individuals in the three item choice dataset by [8] (see S1 Fig). Importantly, we assume that ten individuals do not exhibit a gaze bias, meaning that their choices are independent of the time that they spend looking at each item. To this end, we set the $\gamma$ value of ten randomly selected individuals to 1. We further assume that individuals' gaze is distributed randomly with respect to the values of the items in a choice set. An overview of the generating parameter estimates is given in S3 Fig.

We first instantiate a GLAM model instance using `gb.GLAM()` and then use its `simulate_group` method. This method requires us to specify whether the individuals of the group are either simulated individually (and thereby independent of one another) or as part of a group with hierarchical parameter structure (where the individual model parameters are drawn from a group distribution, see below). For the former, the generating model parameters (indicated in the following as `gen_parameters`) are provided as a dictionary, containing a list of the individual participant values for each model parameter:

```python
import glambox as gb
import numpy as np
glam = gb.GLAM()
no_bias_subjects = np.random.choice(a=gen_parameters
['gamma'].size,
                                    size=10,
                                    replace=False)
gen_parameters['gamma'][no_bias_subjects] = 1
glam.simulate_group(kind='individual',
                    n_individuals=30,
                    n_trials=300,
                    n_items=3,
                    parameters=gen_parameters)
```

As this example is focused on the individual level, we can further create a summary table, describing individuals' response behaviour on three behavioural metrics, using the `aggregate_subject_level_data` function from the `analysis` module. The resulting table contains individuals' mean RT, their probability of choosing the item with the highest item value from a choice set and a behavioural measure of the strength of the association between individuals' gaze allocation and choice behaviour (indicating the mean increase in choice probability for an item that was fixated on longer than the others, after correcting for the influence of the item value on choice behaviour; for further details, see [21]).

```python
from glambox.analysis import aggregate_subject_level_data
subject_data_summary = aggregate_subject_level_data
(data=glam.data,
                                                   n_items=3)
```

**Exploring the behavioural data.** In a first step of our analysis, we explore differences in individuals' response behaviour. To this end, we plot the distributions of individuals' scores on

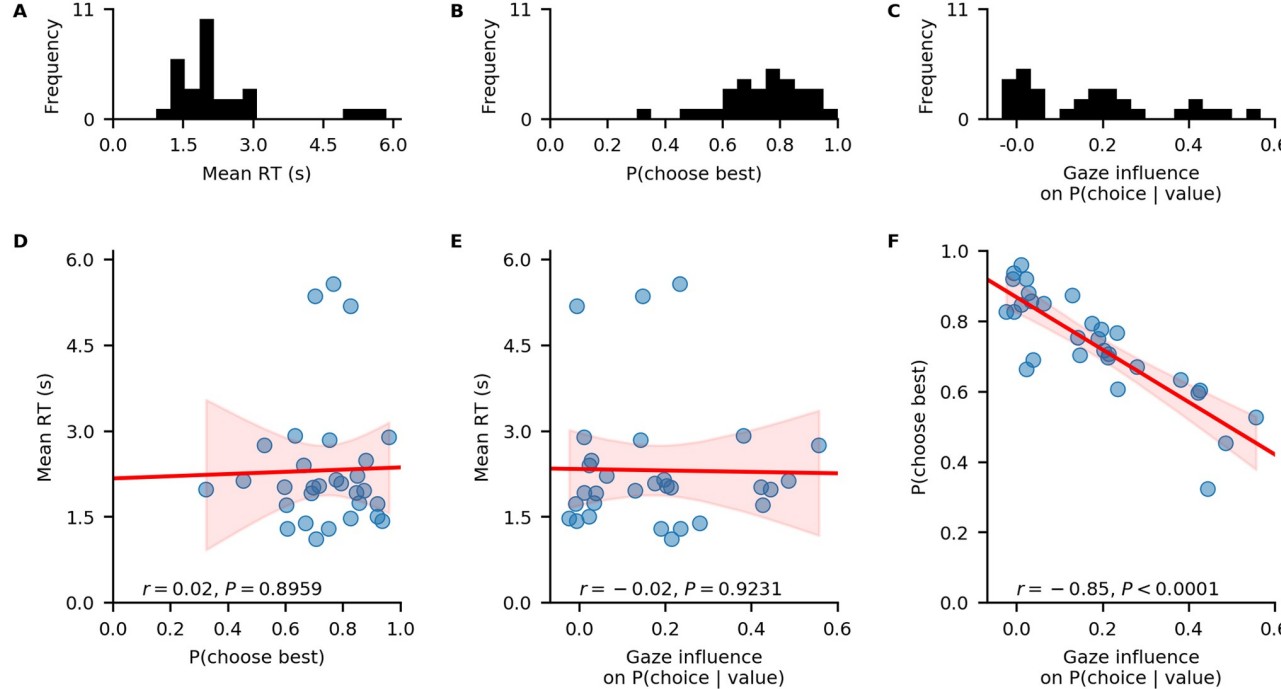

**Fig 3. Individual differences in the data.** A-C: distributions of individuals' mean RT (A), probability of choosing the highest-valued item in a trial (B), and behavioural influence of gaze allocation on choice behaviour (C). D-F: associations between individuals' probability of choosing the highest-valued item and mean RT (D), individuals' behavioural influence of gaze allocation on choice behaviour and their mean RT (E), individuals' behavioural influence of gaze allocation on choice behaviour and their probability of choosing the highest-valued item (F). Red lines indicate linear regression fits with confidence bands surrounding them. Pearson's $r$ coefficients with corresponding $P$-values are reported for each association in D-F.

the three behavioural metrics, and their associations, using the `plot_behaviour_associations` function implemented in the `plots` module:

```
gb.plots.plot_behaviour_associations(data=data)
```

The resulting plot is displayed in Fig 3 and shows that individuals' probability of choosing the best item, as well as the strength of their behavioural association of gaze and choice, are not associated with their mean RT (Fig 3D and 3E). However, individuals' probability of choosing the best item increases with decreasing strength of the behavioural association of gaze and choice (Fig 3F).

**Likelihood-based model comparison.** In a second step of our analysis, we want to test whether the response behaviour of each individual is better described by a decision model with or without gaze bias. To this end, we set up the two GLAM variants:

```
glam_bias = gb.GLAM(data=data)
glam_bias.make_model(kind='individual', name='glam_bias')

glam_nobias = gb.GLAM(data=data)
glam_nobias.make_model(kind='individual', gamma_val=1,
name='glam_nobias')
```

For the GLAM variant without gaze bias mechanism, we use the `gamma_val` argument and set it to a value of 1 (fixing $\gamma$ to 1 for all subjects). We also assign different names to each model with the `name` attribute to better identify them in our subsequent analyses.

Subsequently, we fit both models to the data of each individual and compare their fit by means of the Widely Applicable Information Criterion (WAIC; [34]):

```
glam_bias.fit(method='MCMC',
              tune=5000,
              draws=5000,
              chains=4)

glam_nobias.fit(method='MCMC',
                tune=5000,
                draws=5000,
                chains=4)
```

The fit method defaults to Metropolis-Hastings MCMC sampling (for methodological details, see Methods Section). The `draws` argument sets the number of samples to be drawn. This excludes the tuning (or burn-in) samples, which can be set with the `tune` argument. In addition, the `fit` method accepts the same keyword arguments as the PyMC3 sample function, which it wraps (see the PyMC3 documentation for additional details). The `chains` argument sets the number of MCMC traces (it defaults to four and should be set to at least two, in order to allow convergence diagnostics).

After convergence has been established for all parameter traces (for details on the suggested convergence criteria, see Methods), we perform a model comparison on the individual level, using the `compare_models` function from the `analysis` (see Basic Usage: Comparing model variants):

```
comparison_df = gb.analysis.compare_models(models=[glam_bias,
glam_nobias],

                                           ic='WAIC')
```

The resulting table (shown in Table 2) can be used to identify the best fitting model (indicated by the lowest WAIC score) per individual.

With this comparison, we are able to identify those participants whose response behaviour matches the assumption of gaze-biased evidence accumulation. In particular, we find that we accurately recover whether an individual has a gaze bias or not for 29 out of 30 individuals.

Looking at the individual parameter estimates (defined as MAP of the posterior distributions), we find that the individually fitted $\gamma$ values (Fig 4A) cover a wide range between -0.8 and 1, indicating strong variability in the strength of individuals' gaze bias. We also find that $\gamma$ estimates have a strong negative correlation with individuals' scores on the behavioural gaze bias measure (Fig 4B).

**Out-of-sample prediction.** We have identified those participants whose response behaviour is better described by a GLAM variant with gaze-bias than one without. Yet, this analysis does not indicate whether the GLAM is a good model of individuals' response behaviour on an absolute level. To test this, we perform an out-of-sample prediction exercise.

**Table 2. Output from `compare_models` function for the first two subjects.**

| subject | model | WAIC | pWAIC | dWAIC | weight | SE | dSE | var_warn |
|---------|-------|------|-------|-------|--------|-----|-----|----------|
| 0 | glam_bias | 523.6 | 5.75 | 0 | 0.94 | 50.25 | 0 | 0 |
| 0 | glam_nobias | 645.09 | 3.64 | 121.49 | 0.06 | 44.15 | 23.56 | 0 |
| 1 | glam_bias | 1097.86 | 3.69 | 0 | 1 | 40.32 | 0 | 0 |
| 1 | glam_nobias | 1185.02 | 2.85 | 87.16 | 0 | 38.22 | 18 | 0 |

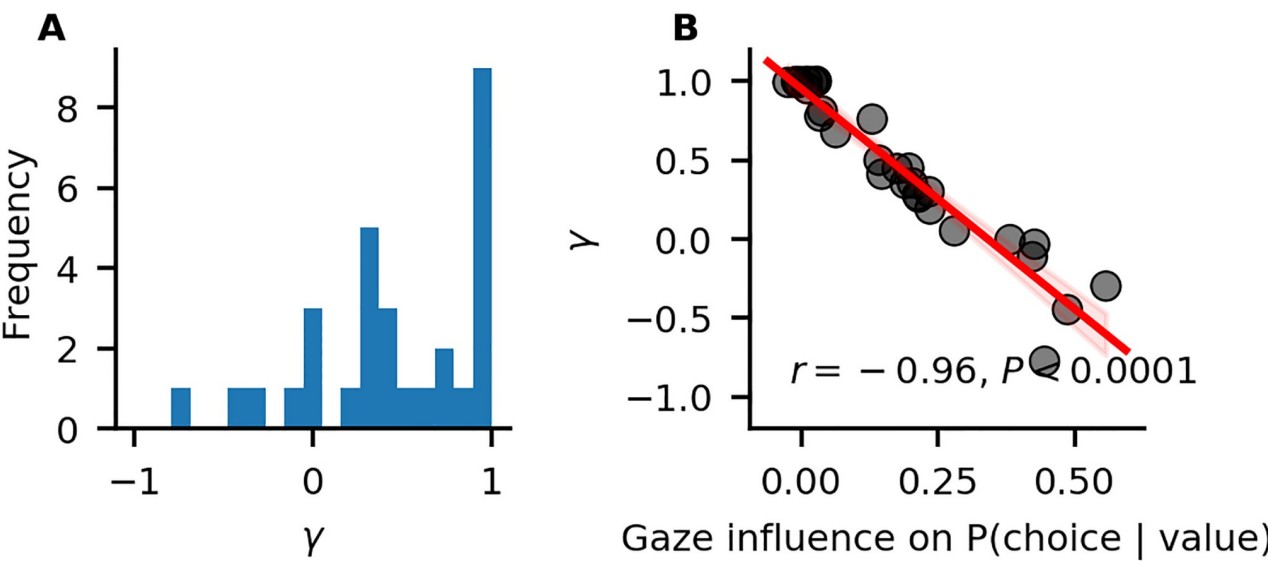

**Fig 4. Individual differences in the strength of the association of gaze allocation and choice behaviour.** A: Distribution of $\gamma$ estimates resulting from the in-sample individual model fits. B: Association of $\gamma$ estimates and individuals' values on the behavioural gaze bias measure. The red line indicates a linear regression fit, with surrounding 95% confidence bands. Pearson's $r$ correlation with $P$-value is given.

We divide the data of each subject into even- and odd-numbered experiment trials and use the data of the even-numbered trials to fit both GLAM variants:

```
glam_bias.exchange_data(data_even)
glam_bias.fit(method='MCMC',
              tune=5000,
              draws=5000,
              chains=4)

glam_nobias.exchange_data(data_even)
glam_nobias.fit(method='MCMC',
              tune=5000,
              draws=5000,
              chains=4)
```

Subsequently, we evaluate the performance of both models in predicting individuals' response behaviour using the MAP estimates and item value and gaze data from the odd-numbered trials. To predict response behaviour for the odd-numbered trials, we use the `predict` method. We repeat every trial 50 times in the prediction (as specified through the `n_repeats` argument) to obtain a stable pattern of predictions:

```
glam_bias.exchange_data(data_odd)
glam_bias.predict(n_repeats=50)

glam_nobias.exchange_data(data_odd)
glam_nobias.predict(n_repeats=50)
```

Lastly, to determine the absolute fit of both model variants to the data, we plot the individually predicted against the individually observed data on all three behavioural metrics. To do this, we use the `plot_individual_fit` function of the `plots` module. This function takes as input the observed data, as well as a list of the predictions of all model variants that ought to be compared. The argument `prediction_labels` specifies the naming used for

each model in the resulting figure. For each model variant, the function creates a row of panels, plotting the observed against the predicted data:

```
from glambox.plots import plot_individual_fit
plot_individual_fit(observed = data_odd,
                    predictions=[glam_bias.prediction,
                                 glam_nobias.prediction],
                    prediction_labels=['gaze-bias',
                                       'no gaze-bias'])
```

The resulting plot is displayed in Fig 5. We find that both model variants perform well in capturing individuals' RTs and probability of choosing the best item (Fig 5A, 5D, 5B and 5E). Importantly, only the GLAM variant with gaze bias is able to also recover the strength of the association between individuals' choice behaviour and gaze allocation (Fig 5C).

**Conclusion.** GLAMbox provides an easy-to-use tool to test the presence (and variability) of gaze biases on the individual level. With GLAMbox, we can easily fit the GLAM to individual participant data, compare different model variants and predict individuals' response behaviour. It also provides a set of analysis functions to explore behavioural differences between individuals and to compare the fit of different model variants to observed response behaviour.

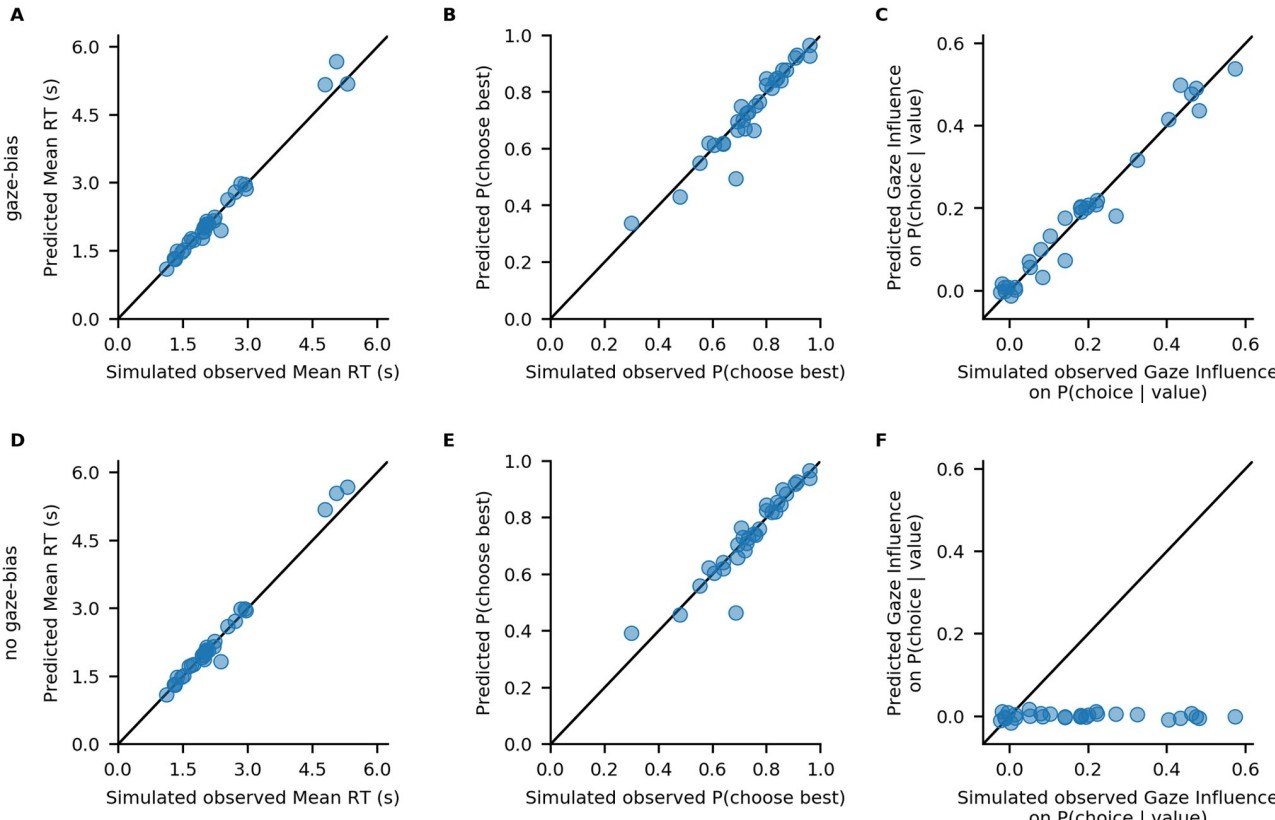

**Fig 5. Out-of-sample model fits.** Comparison of individuals' simulated observed response behaviour with the out-of-sample predictions of a GLAM variant with (A-C) and without gaze bias (D-F): Individuals' mean RT (A, D), probability of choosing the best item (B, E), and influence of gaze allocation on choice probability (C, F). Points indicate individual participant means.

## Example 2: Hierarchical parameter estimation in cases with limited data

In some research settings, the total amount of data one can collect per individual is limited, conflicting with the large amounts of data required to obtain reliable and precise individual parameter estimates from diffusion models [35, 36]. Hierarchical modeling can offer a solution to this problem. Here, each individual's parameter estimates are assumed to be drawn from a group level distribution. Thereby, during parameter estimation, individual parameter estimates are informed by the data of the entire group. This can greatly improve parameter estimation, especially in the face of limited amounts of data [25, 37]. In this example, we will simulate a clinical application setting, in which different patient groups are to be compared on the strengths of their gaze biases, during a simple value-based choice task that includes eye tracking. It is reasonable to assume that the amount of data that can be collected in such a setting is limited on at least two accounts:

1. The number of patients available for the experiment might be low

2. The number of trials that can be performed by each participant might be low, for clinical reasons (e.g., patients feel exhausted more quickly, time to perform tests is limited, etc.)

Therefore, we simulate a dataset with a low number of individuals within each group (between 5 and 15 per group), and a low number of trials per participant (50 trials). We then estimate model parameters in a hierarchical fashion, and compare the group level gaze bias parameter estimates between groups.

**Simulating data.**    We simulate data of three patient groups ($N_1 = 5$, $N_2 = 10$, $N_3 = 15$), with 50 trials per individual, in a simple three item value-based choice task, where participants are instructed to simply choose the item they like the best. These numbers are roughly based on a recent clinical study on the role of the prefrontal cortex in fixation-dependent value representations [13]. Here, the authors found no systematic differences between frontal lobe patients and controls on integration speed or the decision threshold, controlling speed-accuracy trade-offs. Therefore, in our example we only let the gaze bias parameter $\gamma$ differ systematically between the groups, with means of $\gamma_1 = 0.7$ (weak gaze bias), $\gamma_2 = 0.1$ (moderate gaze bias) and $\gamma_3 = -0.5$ (strong gaze bias), respectively. We do not assume any other systematic differences between the groups and sample all other model parameters from the estimates obtained from fitting the model to the data of [8] (for an overview of the generating parameters, see S4 Fig).

Behavioural differences between the three groups are plotted in Fig 6, using the `plot_behaviour_aggregate` function from the `plots` module. Group-level

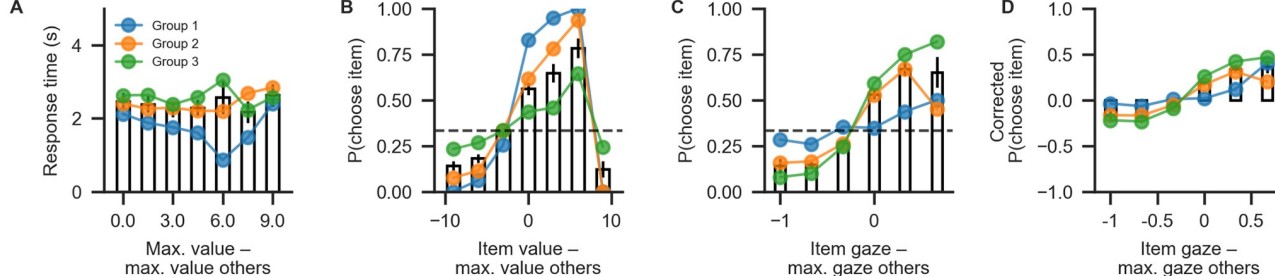

**Fig 6. Aggregate view of the simulated data for Example 2.** (A) Mean RT binned by trial difficulty (the difference between the highest item value in a choice set and the maximum value of all other items). (B) The probability that an item is chosen based on its relative value (the difference of the item's value and the maximum value of all other items in the choice set). (C) The probability of choosing an item based on its relative gaze (the difference between the gaze towards this item and the maximum gaze towards a different item). (D) The probability of choosing an item based on its relative gaze, when correcting for the influence of its value. Bars correspond to the pooled data, while coloured lines indicate individual groups.

summary tables can be created using the `aggregate_group_level_data` from the `analysis` module. Even though the groups only differ in the gaze bias parameter, they also exhibit differences in RT (Group 1 mean ± s.d. = 1.96 ± 0.33 s, Group 2 mean ± s.d. = 2.38 ± 1.4 s; Group 3 mean ± s.d. = 2.59 ± 1.26 ms; Fig 6A) and choice accuracy (Group 1 mean ± s.d. = 0.88 ± 0.06, Group 2 mean ± s.d. = 0.71 ± 0.07, Group 3 mean ± s.d. = 0.50 ± 0.16; Fig 6B). As is to be expected, we can also observe behavioural differences in gaze influence measure (Group 1 mean ± s.d. = 0.08 ± 0.07, Group 2 mean ± s.d. = 0.26 ± 0.11, Group 3 mean ± s.d. = 0.38 ± 0.11; Fig 6C and 6D, where the choices of Group 3 are driven by gaze more than those of the other groups.

**Building the hierarchical model.** When specifying the hierarchical model, we allow all model parameters to differ between the three groups. This way, we will subsequently be able to address the question whether individuals from different groups differ on one or more model parameters (including the gaze bias parameter $\gamma$, which we are mainly interested in here). As for the individual models, we first initialize the model object using the GLAM class and supply it with the behavioural data using the `data` argument. Here, we set the model kind to '`hierarchical`' (in contrast to '`individual`'). Further, we specify that each model parameter can vary between groups (referring to a '`group`' variable in the data):

```
hglam = gb.GLAM(data=data)
hglam.make_model(kind='hierarchical',
                 depends_on = dict(v='group',
                                   gamma='group',
                                   s='group',
                                   tau='group'))
```

In this model, each parameter is set up hierarchically within each group, so that individual estimates are informed by other individuals in that group. If the researcher does not expect group differences on a parameter, this parameter can simply be omitted from the `depends_on` dictionary. The resulting model would then have a hierarchical setup of this parameter across groups, so that individual parameter estimates were informed by all other individuals (not only those in the same group).

**Parameter estimation with MCMC.** After the model is built, the next step is to perform statistical inference over its parameters. As we have done with the individual models, we can use MCMC to approximate the parameters' posterior distributions (see Methods for details). Due to the more complex structure and drastically increased number of parameters, the chains from the hierarchical model usually have higher levels autocorrelation. To still obtain a reasonable number of effective samples [32], we increase the number of tuning- and draw steps:

```
hglam.fit(method='MCMC',
          draws=20000,
          tune=20000,
          chains=4)
```

**Evaluating parameter estimates, interpreting results.** After sampling is finished, and the chains were checked for convergence, we can turn back to the research question: Do the groups differ with respect to their gaze biases? Questions about differences between group-level parameters can be addressed by inspecting their posterior distributions. For example, the probability that the mean $\gamma_{1,\mu}$ for Group 1 is larger than the mean $\gamma_{2,\mu}$ of Group 2 is given by the proportion of posterior samples in which this was the case.

GLAMbox includes a `compare_parameters` function that plots posterior distributions of group level parameters. Additionally, the user can specify a list of comparisons between

groups or conditions. If comparisons are specified, the posterior distributions of their difference and corresponding relevant statistics are added to the figure:

```
from glambox.plots import compare_parameters
parameters = ['v', 'gamma', 's', 'tau']
comparisons = [('group1', 'group2'),
               ('group1', 'group3'),
               ('group2', 'group3')]
compare_parameters(model=hglam,
                   parameters=parameters,
                   comparisons=comparisons)
```

With the resulting plot (Fig 7), the researcher can infer that the groups did not differ with respect to their mean velocity parameters $v_{i,\mu}$ (top row, pairwise comparisons), mean accumulation noise $\sigma_{i,\mu}$ (third row), or scaling parameters $\tau_{i,\mu}$. The groups differ, however, in the strength of their mean gaze bias $\gamma_{i,\mu}$ (second row): All differences between the groups were

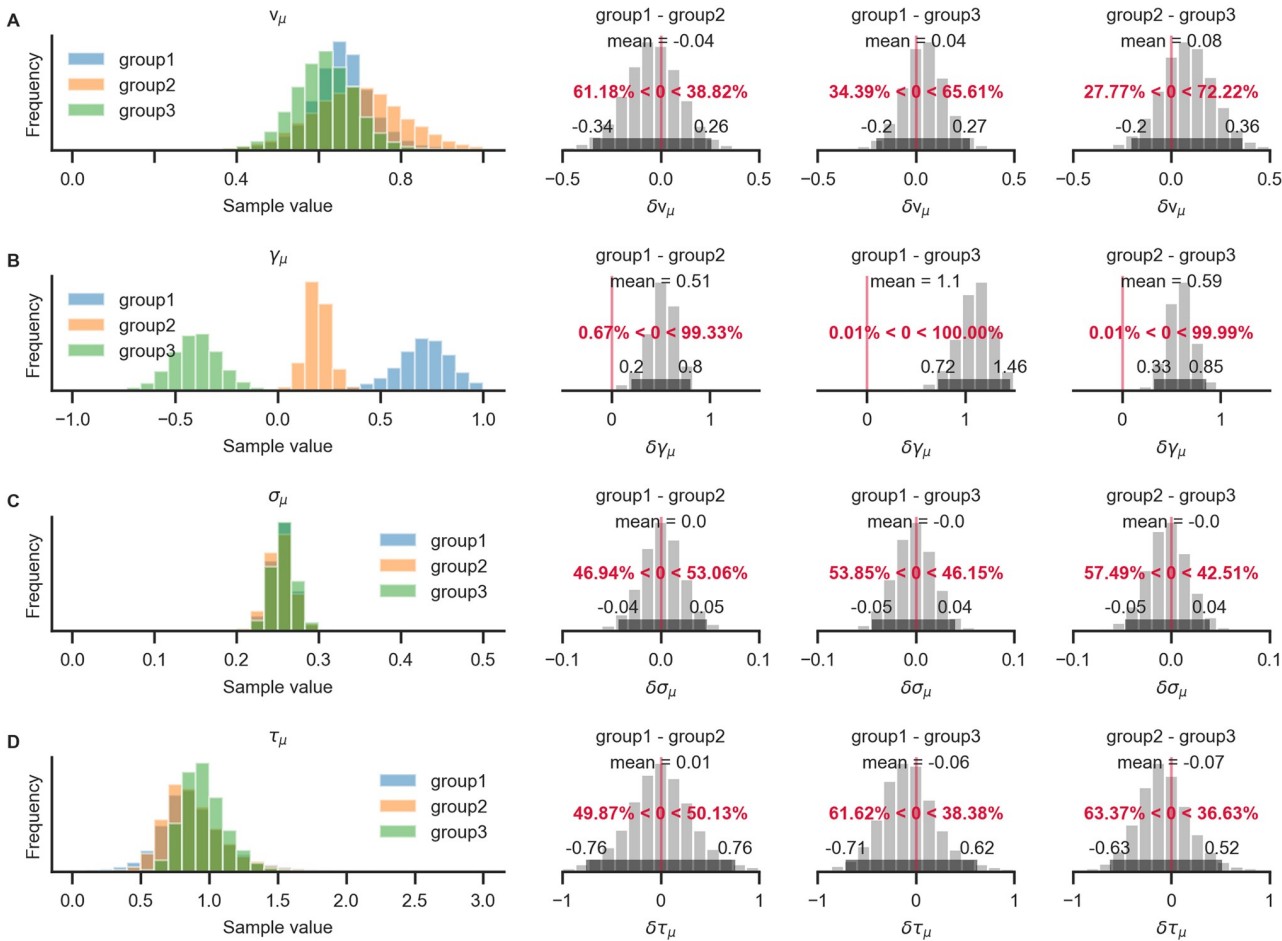

**Fig 7. Pairwise comparison of posterior group-level parameter estimates between groups.** Each row corresponds to one model parameter. The leftmost column shows the estimated posterior distributions for each parameter and group. Pairwise differences between the group posterior distributions are shown in all other columns. For each posterior distribution of the difference, the mean and 95% HPD are indicated, as well as the proportion of samples below and above zero (in red). All three groups differ on the $\gamma$ parameter (row B). No evidence for differences on any of the other model parameters is found (the 95% HPD of the pairwise differences between groups all include zero).

statistically meaningful (as inferred by the fact that the corresponding 95% HPD did not contain zero; second row, columns 2-4).

**Conclusion.** When faced with limited data, GLAMbox allows users to easily build and estimate hierarchical GLAM variants, including conditional dependencies of model parameters. The Bayesian inference framework allows the researcher to answer relevant questions in a straightforward fashion. To this end, GLAMbox provides basic functions for computation and visualization.

### Example 3: Parameter recovery

When performing model-based analyses of behaviour that include the interpretation of parameter estimates, or comparisons of parameter estimates between groups or conditions, the researcher should be confident that the model's parameters are actually identifiable. In particular, the researcher needs to be confident that the set of estimated parameters unambiguously describes the observed data better than any other set of parameters. A straightforward way of testing this is to perform a parameter recovery: The general intuition of a parameter recovery analysis is to first generate a synthetic dataset from a model using a set of known parameters, and then fitting the model to the synthetic data. Finally, the estimated parameters can be compared to the known generating parameters. If they match to a satisfying degree, the parameters were recovered successfully. Previous analyses have already indicated that the GLAM's parameters can be recovered to a satisfying degree [21]. Yet, the ability to identify a given set of parameters always depends on the specific features of a given dataset. The most obvious feature of a dataset that influences recoverability of model parameters is the number of data points included. Usually this quantity refers to the number of trials that participants performed. For hierarchical models, the precision of group-level estimates also depends on the number of individuals per group. Additional features that vary between datasets and that could influence parameter estimation are the observed distribution of gaze, the distribution of item values or the number of items in each trial. For this reason, it is recommended to test whether the estimated parameters of a model can be recovered in the context of a specific dataset. slac To demonstrate the procedure of a basic parameter recovery analysis using GLAMbox, suppose we have collected and loaded a dataset called `data`. In the first step, we perform parameter estimation as in the previous examples:

```
glam = gb.GLAM(data=data)
glam.make_model(kind='individual')
glam.fit(method='MCMC',
        draws=5000,
        tune=5000,
        chains=4)
```

The next step is to create a synthetic, model-generated dataset using the model parameters estimated from the empirical data, together with the empirically observed stimulus and gaze data using the `predict` method. Setting `n_repeats` to 1 results in a dataset of the same size as the observed one:

```
glam.predict(n_repeats=1)
synthetic = glam.prediction
```

The synthetic dataset should resemble the empirically observed data closely. If there are major discrepancies between the synthetic and observed data, this indicates that GLAM might not be a good candidate model for the data at hand. Next, we create a new model instance, attach the synthetic data, build a model and re-estimate its parameters:

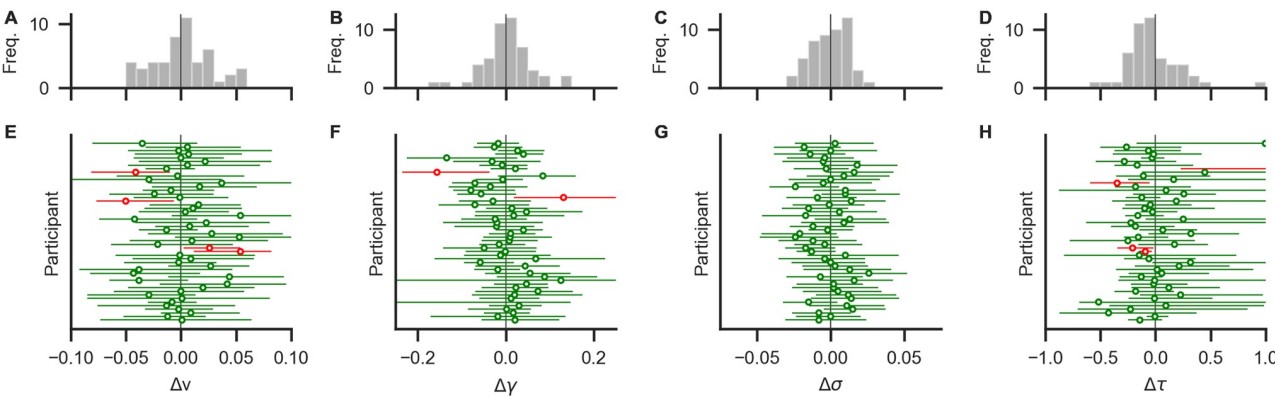

**Fig 8. Results from a basic parameter recovery.** The lower row (E-H) shows deviations between known generating parameter values and recovered MAP estimates (circles) and their 95% HPDs (horizontal error bars) for each participant. Green (red) colour indicates that the true value is within (outside) the 95% HPD. Most parameters were recovered with small deviations. Panels A-D show distributions of deviations across individuals. Distributions are mostly centered around zero, indicating no systematic under- or overestimation (bias) across individuals.

```
glam_rec = gb.GLAM(data=synthetic)
glam_rec.make_model(kind='individual')
glam_rec.fit(method='MCMC',
             draws=5000,
             tune=5000,
             chains=4)
```

Finally, the recovered and generating parameters can be compared. If the recovered parameters do not match the generating parameters, the parameters cannot be identified given this specific dataset. In this case, parameter estimates should not be interpreted.

If, on the other hand, generating and recovered parameters do align, the parameters have been recovered successfully. This indicates that the model's parameters can be identified unambiguously given the general characteristics of the dataset and thereby increases confidence that the parameters obtained from the empirical data are valid and can be interpreted.

Here, all parameters could be recovered as illustrated in Fig 8. For most individuals, the MAP estimates and their 95% HPDs are close to the known generating parameters. Across individuals, no systematic biases in the estimation can be identified.

**Conclusion.** In this example, we demonstrated how to perform a basic parameter recovery for a given dataset. When successful, this increases confidence that the parameters can be identified with the given dataset.

## Discussion

Researchers have recently started to systematically investigate the role of visual gaze in the decision making process. By now, it is established that eye movements do not merely serve to sample information that is then processed independently to produce a choice, but that they are actively involved in the construction of preferences [2, 4, 6–8, 10, 14, 15, 21, 38]. The dominant theoretical perspective is that evidence accumulation in favor of each option is modulated by gaze allocation, so that accumulation for non-fixated options is attenuated. This mechanism is formally specified in various models of gaze-dependent decision making, such as the attentional Drift Diffusion Model (aDDM; [7, 8]) and the conceptually related Gaze-weighted Linear Accumulator Model (GLAM; [21]). In contrast to analyses based on behavioural and eye

tracking data alone, these models can act as analytical tools that enable researchers to address questions regarding specific mechanisms in the decision process, like the gaze bias.

The goal of GLAM is to provide a model-based estimate of the gaze bias on the level of an individual (as indicated by GLAM's $\gamma$ parameter), in choice situations involving more than two choice alternatives. To estimate the gaze bias, GLAM describes the decision process in the form of a linear stochastic race and aggregates over the specific sequence of fixations during the decision process (by only utilizing the fraction of the decision time that each item was looked at). These two characteristics distinguish the GLAM from other existing approaches of obtaining an estimate of individuals' gaze bias:

First, the GLAM is focused on quantifying the gaze bias on the individual level. It does not capture dynamics of the decision process on the level of single fixations. If these fine-grained dynamics are of interest to the researcher, the aDDM can be used. Here, the fixation-dependent changes in evidence accumulation rates throughout the trial are not averaged out. Keeping this level of detail, however, comes at a cost: Fitting the aDDM relies on extensive model simulations (including a simulation of the fixation process; for a more detailed discussion see [21]). The GLAM, on the other hand, aggregates over the fixation-dependent changes in the accumulator's drift rate in order to simplify the estimation process of the gaze bias.

Second, the GLAM directly applies to choice situations involving more than two choice alternatives. While the GLAM has been shown to also capture individuals' gaze bias and choice behaviour well in two-alternative choice situations [21], there exist other computational approaches that can estimate the gaze bias of an individual in binary decisions: If response times are of interest to the researcher, the gaze bias can be estimated in the form of a gaze-weighted DDM (see for example [2, 18]). Similar to the GLAM, this approach also aggregates over the dynamics of the fixation process within a trial, by only utilizing the fraction of trial time that each item was looked at. In contrast to the GLAM, however, gaze-weighted DDM approaches describe the decision process in the form of a single accumulator that evolves between two decision bounds (each representing one of the two choice alternatives). For two-alternative choice scenarios, where response times are not of interest to the researcher, Smith and colleagues [39] proposed a method of estimating the aDDM gaze-bias parameter through a random utility model. Here, the gaze bias can be estimated in a simple logit model.

Even though the advantages of applying these types of models are apparent, their use is often limited by their complexity and the high cost of implementing, validating and optimizing them. Further, there are only few off-the-shelf solutions researchers can turn to, if they want to perform model-based analyses of gaze-dependent choice data, particularly for choice settings involving more than two alternatives.

With GLAMbox, we present a Python-based toolbox, built on top of PyMC3, that allows researchers to perform model-based analyses of gaze-bias effects in decision making easily. We have provided step-by-step instructions and code to perform essential modeling analyses using the GLAM. These entail application of the GLAM to individual and group-level data, specification of parameter dependencies for both within- and between-subject designs, (hierarchical) Bayesian parameter estimation, comparisons between multiple model variants, out-of-sample prediction of choice and RT data, data visualization, Bayesian comparison of posterior parameter estimates between conditions, and parameter recovery. We hope that GLAMbox will make studying the association between gaze allocation and choice behaviour more accessible. We also hope that the resulting findings will ultimately help us better understand this association, its inter-individual variability and link to brain activity.

## Supporting information

**S1 Fig. Distribution of individual parameter estimates from four datasets analysed in Thomas et al. (2019).** The top row contains distributions of parameter estimates across datasets. Subsequent rows show distributions per dataset: Krajbich et al. (2010; blue), Krajbich & Rangel (2011; orange), Experiment 2 from Folke et al. (2017; green) and Experiment 1 from Tavares et al. (2017; red).
(TIFF)

**S2 Fig. Illustration of hyperpriors.** Different hyperpriors based on group-averaged parameter values were obtained from fitting the model to four different datasets (Folke et al., 2017; Krajbich et al., 2010; Krajbich & Rangel, 2011; Tavares et al., 2017; see S1 Table and S1 Fig). Panels show prior distributions on group level mean (upper row) and standard deviation (lower row) for each model parameter (columns; from left to right: $v$, $\gamma$, $\sigma$, $\tau$). Observed group level estimates from the four datasets are indicated as red ticks in each panel. Blue, orange and green lines represent prior distributions with increasing levels of vagueness $f$. They are constructed as normal distributions with mean equal to the mean of the observed group level parameters across datasets (M), and standard deviation equal to $f$ times the observed standard deviation across datasets (SD). Higher values of $f$ correspond to wider, less informative priors. Prior distributions are further bounded between sensible limits. The user can specify the factor $f$ during specification of hierarchical models. By default, hyperpriors with $f = 10$ (orange lines) are used.
(TIFF)

**S3 Fig. Distribution of individual generating GLAM parameters of Example 1.** Colours indicate whether a subject was simulated with or without gaze bias.
(TIFF)

**S4 Fig. Distributions of data-generating parameters for the three groups in Example 2.** The top row shows distributions pooled across groups. The bottom three rows show distributions per group. Note that the groups do not differ systematically with respect to the velocity parameter $v$, the noise parameter $\sigma$, or the scaling parameter $\tau$ (first, second and last column; even though there is some variability between individuals). The groups differ, however, on the gaze bias parameter $\gamma$ (third column): Group 1 only has a weak gaze bias (large $\gamma$), group 2 has a medium strong gaze bias (smaller $\gamma$), and group 3 has a very strong gaze bias (even smaller, negative $\gamma$).
(TIFF)

**S1 Table. Description of individual parameter estimates from four datasets analysed in Thomas et al. (2019).** The datasets are originally from Folke et al., 2017 (Experiment 2); Krajbich et al., 2010; Krajbich & Rangel, 2011 and Tavares et al., 2017 (Experiment 1).
(PDF)

## Author Contributions

**Conceptualization:** Felix Molter, Armin W. Thomas, Hauke R. Heekeren, Peter N. C. Mohr.

**Formal analysis:** Felix Molter, Armin W. Thomas.

**Software:** Felix Molter, Armin W. Thomas.

**Supervision:** Hauke R. Heekeren, Peter N. C. Mohr.

**Visualization:** Felix Molter, Armin W. Thomas.

**Writing – original draft:** Felix Molter, Armin W. Thomas.

**Writing – review & editing:** Felix Molter, Armin W. Thomas, Hauke R. Heekeren, Peter N. C. Mohr.

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
