## [Decision Letter · Decision Letter 0]

26 Sep 2019

PONE-D-19-24023

GLAMbox: A Python toolbox for investigating the association between gaze allocation and decision behaviour

PLOS ONE

Dear Prof. Dr. Mohr,

Thank you for submitting your manuscript to PLOS ONE. After careful consideration, we feel that it has merit but does not fully meet PLOS ONE’s publication criteria as it currently stands. Therefore, we invite you to submit a revised version of the manuscript that addresses the points raised during the review process.

All three reviewers were enthusiastic about the paper and toolbox. The primary recommendations involve clarifying some of the parameter choices and their impact on the results and interpretation. The reviewers also highlight the importance of contextualizing the approach and comparing/contrasting with previous work. In addition, note that Reviewers 2 and 3 both had trouble using the toolbox, so it would be important to update documentation and examples in your revision. 

We would appreciate receiving your revised manuscript by Nov 10 2019 11:59PM. To enhance the reproducibility of your results, we recommend that if applicable you deposit your laboratory protocols in protocols.io, where a protocol can be assigned its own identifier (DOI) such that it can be cited independently in the future. For instructions see: http://journals.plos.org/plosone/s/submission-guidelines#loc-laboratory-protocols

We look forward to receiving your revised manuscript.

Kind regards,

David V. Smith, Ph.D.

Academic Editor

PLOS ONE

Journal Requirements:

The Junior Professorship of P.N.C.M. as well as the associated Dahlem International Network Junior Research Group Neuroeconomics is supported by Freie Universit ¨at Berlin within the Excellence Initiative of the German Research Foundation (DFG). Further support for P.N.C.M. is provided by the WZB Berlin Social Science Center. F.M. is supported by the International Max Planck Research School on the Life Course (LIFE). The funders had no role in study design, data collection and analysis, decision to publish or preparation of the manuscript

Reviewers' comments:

Reviewer's Responses to Questions

**Comments to the Author**

1. Is the manuscript technically sound, and do the data support the conclusions?

Reviewer #1: Yes

Reviewer #2: Yes

Reviewer #3: Yes

2. Has the statistical analysis been performed appropriately and rigorously? 

Reviewer #1: Yes

Reviewer #2: Yes

Reviewer #3: Yes

3. Have the authors made all data underlying the findings in their manuscript fully available?

Reviewer #1: Yes

Reviewer #2: Yes

Reviewer #3: Yes

4. Is the manuscript presented in an intelligible fashion and written in standard English?

Reviewer #1: Yes

Reviewer #2: Yes

Reviewer #3: Yes

5. Review Comments to the Author

Reviewer #1: This is a very interesting study that introduces the gaze-weighted linear accumulator model (GLAM) and validates a new toolbox for fitting the model in Python to understand the association between gaze bias and decision making. In this manuscript, the authors tried to validate the GLAM and the toolbox in three different cases: individual-level parameter estimation, group-level parameter estimation using hierarchical Bayesian method, and parameter recovery. The authors show convincing and converging evidence that the GLAM performs better than a model without reflecting gaze bias in value-based decisions and explain how to use the toolbox to fit GLAM in Python. I believe the manuscript is good to be considered publication in PLOS ONE with a revision. Here, I summarize several points that I was not clear or had questions while reading it through:

1. It was not quite clear what the general speed parameter (parameter v in Eq 5, page 4) in the model captures. What does the general speed exactly mean? Does it capture speed-accuracy tradeoff as boundary parameter in DDM? It seems like Eq 6 captures accuracy-speed tradeoff using the speed parameter, but it is not quite clear for me. If the authors could provide more about this parameter, it would be better for readers to understand the parameter better.

2. Is Eq 4 correct? Parentheses are missing after “exp”?

3. In Example 1, the authors collected liking scores after choice and used the liking scores to identify the higher value items or the best items. However, choice-induced preference literature has shown that choices not only reveal preferences, but also shape preferences. Thus, chosen items might have higher liking scores than non-chosen items not only because participants liked them but also because they chose them. Is there any way to rule out this issue? Or, do the authors expect different or the results if liking scores are measured in advance and test the model?

4. The “glam_bias.fit” and “glam_nobias.fit” lines in page 10 and page 11 do not have “chains” attribute (which is 4 in default), but the authors’ suggestion for model convergence is 2 in the main text. I found that the script in Github includes chains parameter in the script. Including the ‘chains attribute in the script examples in the manuscript would help readers more intuitively.

5. The authors mention about the results of model comparison test result in page 11. Without output figure or table, it was not quite easy to understand what the results look like. The Github script did not include model comparison results. Adding the model comparison result table in the revised manuscript would help readers.

6. Overall, I felt that the captions of figures are too long and redundant with the main text. I think it would be better to explain more in the main text and shorten the caption of figures.

7. I could not install the toolbox using pip or conda. If the authors could make it available (or at least inform how to install on a local computer), it would help researchers access the toolbox.

Minor points

8. I am not sure this is a technical issue or not, but the figures were not clear. Some letters were broken too. Please check the clarity of the figures.

9. The number label in page 5 for “individual parameter estimation details” seems quite abrupt. Please drop the numbers.

Reviewer #2: Summary: this paper provides an overview of how to use the authors’ toolbox to measure individual and group differences in the extent to which gaze information influences decision-making. The GLAMbox approach was first introduced in an empirical paper published this year (Thomas et al., 2019), and the current paper expands upon the method to enable other researchers to use it in an informed way. This new method is useful for researchers who use eye tracking as a tool to understand decision-making, and the GLAMbox adds a contribution to the field as a whole. Past research has focused on one overall discount value for unattended information, whereas this allows fitting of individual differences. Moreover, it seems to be a more efficient implementation than past work, which makes it more accessible. The authors are thorough in both describing model-fitting as well as parameter recovery to promote best practices. I think that a few clarifications and additions could make this paper stronger:

1. It would be helpful in the introduction and/or discussion to explain more how different individual gaze biases might arise (familiarity with items, more goal-driven approach, etc.). There is not one obvious reason, but I think some discussion of why this is important is useful. For example, Smith & Krajbich (2018) discuss “tunnel vision” as one possible mechanism.

2. Section 0.0.1 “Individual parameter estimation details” says that the ranges chosen were derived from “sensible limits based on previous applications” (line 118). It would be helpful to have more discussion of how these sensible limits are arrived at, whether they will apply broadly to all data sets, or how to determine appropriate ranges for one’s own data including theoretical constraints. Furthermore, the aDDM that this method seems to draw its inspiration from, uses a discount range for attention of [0-1] (Krajbich et al., 2010; Krajbich et al., 2015). However, the authors here use a range including large negative values (-10) up to 1. I think it’s important to explain why negative values are used, how to interpret them (active forgetting or leaky accumulation?) and to provide a theoretical justification for their inclusion here given the context of previous literature.

3. The GLAM is explained in an option-wise manner. Given that recent research has shown that some individuals compare options with multiple attributes in an attribute-wise manner, would there be a way to incorporate attribute-wise comparisons into the GLAM? This may be outside the scope of the paper, but if there is a relatively easy way to implement it, that could be worth including.

4. Figure 4 shows a strong correlation between gamma and the behavioral gaze bias. This is a good confirmation, but the behavioral measure of gaze-choice association (lines 242-246) is only very briefly mentioned. If they are so highly correlated, what does gamma add beyond the behavioral gaze bias measure? Is its main advantage including it as part of the full model estimation process?

5. Example 2: does it make sense to expect similar parameter ranges for patients compared to a young, healthy sample? Parameters such as noise might be higher and drift rate might be slower. I don’t think that this should affect the gaze bias estimation, but it might affect which parameters are used to constrain hierarchical estimation.

6. In Fig. 6, choice difficulty is defined as the highest value is compared with the average of the other values. However, I think a choice would be more difficult if the second highest value were quite similar to the highest value, regardless of the lower value options. For example, a choice with two similarly high value options and one very low value option would be harder than one with one high value option and 2 medium value options, but both would be similar difficulty by the metric currently used. Is there a reason this is favored over comparing the best and next best options?

7. A different number of draws and burn in samples are used in model-fitting from Example 1 to Example 2; is there a reason for this? Perhaps briefly explain why if it is relevant to users.

Small clarifications/phrasing corrections:

• In the abstract, I would rephrase the middle sentence beginning with “However, only few decision models exist…” to something like, “However, few decision models exist that enable a straightforward characterization of the gaze-choice association at the individual level…”

• In the introduction line 4, “It was repeatedly shown” should be changed to “It has been repeatedly shown”

• Line 66, “i” is not explained. It can be inferred that it indexes each item, but it should be explicitly mentioned.

• Figure 1 and equation 2 (lines 76-77). What does the “maximum of all other decision signals mean”? The highest average absolute decision signal among the item options? My interpretation is that you are subtracting the highest value option from all others as a sort of normalization, but this isn’t quite coming through clearly.

o Line 76, equation 2. What does J represent? From reading the empirical paper using the same method, Thomas et al., 2019, it sounds like J represents the set of all items, but it should also be defined in this paper.

• Figure 1e is above panel d in a way that violates expectations of reading/processing material, and I think it would be clearer if the panel positions for d and e were switched (even though I understand it was likely put there for design reasons).

• Figure 3 flips the orientation of the axes between D, E, and F so that the same variables are on the x versus y axes, which makes it harder to process them all at once. It would better fit your description for “gaze influence on choice” to be on the x-axis in F. I realize that these are non-directional correlations and that the axes may be flipped to better align with the above histograms, but I find it harder to parse this way (instead of just including the histogram distributions with their own separate x-axis labels).

• Figure 5, I might put “simulated observed” instead of just “observed” on the x-axis to make sure that readers don’t get confused and think that the data is actual raw data rather than data simulated from inputted parameters. Alternatively you could mention it in the figure caption.

Reviewer #3: This study presents a python toolbox to fit parameters for the authors’ gaze-weighted linear accumulator model, capitalizing on python’s Bayesian package PyMC3. Fitting a DDM is quite computationally complex, with many researchers who are interested in the theory perhaps not having the skills required to write their own estimation code. Moreover, it is typically a very time-consuming process to fit these kinds of models, so a faster methods are always welcome additions. They use a race model which can handle non-binary choices will help better approximate real-world settings.

I really like that they have included parameter recovery into their toolbox. In addition, doing model comparison with and without the gaze bias parameter is nice – particularly as it can help other researchers understand under which situations gaze is and is not important. Some guidance on what to do to compare multiple conditions/tasks would be a nice feature. In addition, I think the github documentation needs more details and guidance (e.g., simply to tell the reader to use Jupyter to open the readme). In addition, I ran into some errors using the code, which could have been the result of poor documentation.

I have the following suggestions/issues:

I would like to point the authors to Smith, Krajbich, and Webb (Estimating the dynamic role of attention via random utility – 2019) which estimates aDDM’s theta parameter using a very fast and simple regression method, which seems relevant to their work.

In addition, it would be nice to see a discussion/comparison of this to other race models (an unacquainted reader may incorrectly believe that theirs is the first race model to vit ddm-eqsue parameters upon reading their introduction), as well as a discussion of the drawbacks of race models relative to more traditional aDDM methods.

Although this reviewer is familiar with the authors’ previous work on the GLAM model, it may be useful to have a section with more comprehensive introduction to the model/theory and comparison to similar models like the aDDM (subject to editorial guidance – I am not sure what is appropriate).

A much more extensive readme file and instructions should be included. For example, this reviewer know that the examples/readme are to be opened in jupyter, but some (nay, many – esp. those searching for a toolbox rather than writing their own code) may not. A basic guide for others would be helpful.

When I attempted to run the parameter recovery exercise, I received an error originating in glam.fit (AttributeError: Can't pickle local object 'make_subject_model.<locals>.lda_logp'), but don’t know whether that was my poor execution or a problem in the code.

 </locals>

6. PLOS authors have the option to publish the peer review history of their article (what does this mean?). If published, this will include your full peer review and any attached files.

Reviewer #1: No

Reviewer #2: No

Reviewer #3: No

---

## [Author Response · Author response to Decision Letter 0]

12 Nov 2019

Please find attached the document "Response to Reviewers.pdf".

---

## [Decision Letter · Decision Letter 1]

27 Nov 2019

GLAMbox: A Python toolbox for investigating the association between gaze allocation and decision behaviour

PONE-D-19-24023R1

Dear Dr. Mohr,

We are pleased to inform you that your manuscript has been judged scientifically suitable for publication and will be formally accepted for publication once it complies with all outstanding technical requirements.

With kind regards,

David V. Smith, Ph.D.

Academic Editor

PLOS ONE

Additional Editor Comments (optional):

Two out of three of the original reviewers were able to re-review the manuscript. Both reviewers agreed that their original comments had been addressed. Both reviewers also made some additional (minor) suggestions that can be considered in the post-acceptance stage of production. These suggestions were primarily tied to clarifications regarding the model parameters.

Reviewers' comments:

Reviewer's Responses to Questions

**Comments to the Author**

1. If the authors have adequately addressed your comments raised in a previous round of review and you feel that this manuscript is now acceptable for publication, you may indicate that here to bypass the “Comments to the Author” section, enter your conflict of interest statement in the “Confidential to Editor” section, and submit your "Accept" recommendation.

Reviewer #1: All comments have been addressed

Reviewer #2: (No Response)

2. Is the manuscript technically sound, and do the data support the conclusions?

Reviewer #1: Yes

Reviewer #2: Yes

3. Has the statistical analysis been performed appropriately and rigorously? 

Reviewer #1: Yes

Reviewer #2: Yes

4. Have the authors made all data underlying the findings in their manuscript fully available?

Reviewer #1: Yes

Reviewer #2: Yes

5. Is the manuscript presented in an intelligible fashion and written in standard English?

Reviewer #1: Yes

Reviewer #2: Yes

6. Review Comments to the Author

Reviewer #1: (No Response)

Reviewer #2: All of my main concerns have been addressed. I have a few further minor suggestions/clarifications that the authors may consider but don’t require another round of review.

1) The range for most parameters was expanded in the revision, but the range on gamma was narrowed from (-10,1) to (-2,1). I wasn’t sure if the previous range was overly large, or if this was a typo. Looking at the prior data set ranges it seems like this correction is appropriate, but I wanted to flag it since there wasn’t a clear reason.

2) In example 1, (lines 295-297; 355-356) the authors simulate gaze data assuming it is randomly distributed with respect to value, which I agree is the most neutral way to simulate the data. However, there is literature suggesting a bi-directional interaction between attention and value (attention drives value accumulation, but reward associations can also capture gaze). Simply adding a sentence clarifying that the negative relationship between gaze bias and probability of choosing the higher valued options will hold even if the assumption of completely random gaze is relaxed (as confirmed in the authors’ 2019 empirical paper using GLAMbox), but that the relationship might be weaker (ie: if gaze is drawn to higher value options which research suggests is the case) would help acknowledge the bi-directional nature of attention and value.

Small wording corrections:

Line 152-153, “with smaller values producing slower and more accurate responses” I would add “, respectively” to make it clear that smaller values of v produce slower responses and smaller values of sigma produce more accurate responses rather than both serving both roles.

Line 157 “which” should be “with”

Line 305, The paper simulation example doesn't include the code “value_range=(1, 10))”, although it is in the online documentation. I would add it to the paper, as I was confused when reading as to how that information was simulated without being entered into the function.

7. PLOS authors have the option to publish the peer review history of their article (what does this mean?). If published, this will include your full peer review and any attached files.

Reviewer #1: No

Reviewer #2: No

---

## [Editor Report · Acceptance letter]

6 Dec 2019

PONE-D-19-24023R1 

GLAMbox: A Python toolbox for investigating the association between gaze allocation and decision behaviour 

Dear Dr. Mohr:

I am pleased to inform you that your manuscript has been deemed suitable for publication in PLOS ONE. Congratulations! Your manuscript is now with our production department. 

With kind regards,

on behalf of

Dr. David V. Smith 

Academic Editor

PLOS ONE